# A thorough review of the May 5, 1990 earthquake (southern Italy): constraints from macroseismology and insights from hydrology

Andrea Antonucci<sup>1</sup>, Corrado Castellano<sup>2</sup>, Luigi Cucci<sup>2</sup>, Andrea Tertulliani<sup>2</sup>

<sup>1</sup>Istituto Nazionale di Geofisica e Vulcanologia (INGV), Milano, Italy <sup>2</sup>Istituto Nazionale di Geofisica e Vulcanologia (INGV), Roma, Italy

Correspondence to: Andrea Antonucci (andrea.antonucci@ingv.it)

Abstract. The May 5, 1990 Potenza earthquake (Mw 5.77) was a significant event for southern Italy, despite its moderate magnitude and limited damage. Previous macroseismic studies of this earthquake contained inconsistent and often exaggerated intensity values, particularly in areas far from the epicentre. Our analysis reveals that some overestimated intensities were caused by the overlapping damage patterns from previous earthquakes, due to the tendency to emphasize pre-existing or unrepaired damage, or attribute them to the most recent earthquake. In this respect, we re-evaluate all available data from original sources and compile a new and robust dataset comprising 1393 intensity values, assessed using both MCS and EMS-98 scales. This updated dataset shows a general decrease in higher intensity values compared to previous assessments, especially within 150 km of the epicentre. We also identify new data sources and remove unreliable entries. Recalculated macroseismic epicentres are in agreement with the instrumental estimate (i.e., 7.3 km using MCS data), while macroseismic magnitudes (Mw 5.05-5.19) result lower than the instrumental one. Additionally, we collect extensive observations of seismically-induced hydrological changes. These hydrological effects provide independent magnitude estimates ranging from M 4.9 to 5.7 for liquefaction and M 5.2 for streamflow responses. This comprehensive re-evaluation significantly enhances the accuracy and usefulness of the macroseismic and environmental data for future seismological research.

#### 1 Introduction

Macroseismic intensity is a key parameter for seismic hazard assessments and for many seismological approaches that analyse the impact of an earthquake on a given area. A fundamental aspect is the compilation of homogeneous seismic catalogues that maintain "parametric" continuity between macroseismic datasets related to pre-instrumental and modern earthquakes. In particular, seismic catalogues are specifically developed to meet these needs: to ensure a continuous and homogeneous dataset and to allow processing of the available earthquake parameters (i.e., epicentral location, intensity, and magnitude) over large time windows. In Italy, the current Italian Parametric Earthquake Catalogue CPTI15 (Rovida et al. 2020; 2022) covers a period of more than 1,000 years. It contains about 5,000 events, of which over 3,200 are associated with macroseismic intensity data (MDPs) derived from the harmonization and parametrization of the macroseismic intensity data collected in the Italian Macroseismic Database DBMI15 (Locati et al., 2022). Many empirical relationships between parameters of seismological interest are based on macroseismic intensity, for example, to derive Intensity Prediction Equations (IPEs) which describe the ground shaking in terms of macroseismic intensity at a given site (among others Bakun and Scotti 2006; Bindi et al., 2011; Allen et al., 2012; Gomez-Capera et al., 2024; Lolli et al., 2024), and Ground-Motion to Intensity Conversion Equations GMICEs (i.e., Gomez-Capera et al., 2020; Oliveti et al., 2022) or even to calculate macroseismic magnitude of historical events (i.e., Di Maro and Tertulliani, 1990; Gasperini et al., 1999; 2010; Provost and Scotti, 2020; D'Amico et al., 2025). To make these elaborations reliable, intensity values must be as trustworthy as possible, that is, based on a robust and homogeneous body of information. However, due to the nature of intensity itself (e.g., ordinal, discrete and range-limited), datasets often suffer from critical issues and implicit uncertainties, especially for historical earthquakes. These problems have been widely addressed in the literature (i.e., Bakun et al., 2011; Graziani et al., 2015; Hough and Martin, 2021; Vannucci et al., 2021; Del Mese et al., 2023; Antonucci et al., 2025). To this end, recent Italian studies have focused on reassessing the intensity distribution for some events, resulting in a more complete, consistent and comprehensive datasets (i.e., Tertulliani and https://doi.org/10.5194/egusphere-2025-5343 Preprint. Discussion started: 7 November 2025 © Author(s) 2025. CC BY 4.0 License.

Castellano, 2024; Tertulliani et al., 2025) with intensity data provided both in MCS (Sieberg, 1932) and in the EMS-98 scale (Grünthal, 1998). However, some Italian earthquakes still present highly incoherent and inconsistent macroseismic datasets that require in-depth investigation to prevent inaccuracies in future seismological analysis. For this reason, revisions are necessary to reexamine the event's basic data, eliminate sources of uncertainty and misunderstanding, recover new and reliable information, and assign new intensity values based on consistent criteria (Tertulliani et al., 2025). One such case is the earthquake that occurred on May 5, 1990, in southern Italy, with Mw 5.77 according to the CPTI15, which is the focus of this study. Despite its moderate magnitude, it is important to correctly evaluate the effects produced at the sites, since the 1990 earthquake was one of the strongest in a region with relatively low seismicity, and its impact on the territory is of fundamental importance for regional seismic hazard studies. In this work, we first analyse the two macroseismic studies carried out so far on the May 5, 1990 earthquake (Gasparini et al., 1991; Tertulliani et al., 1992) and highlight their intrinsic weaknesses. Then, we provide a new dataset consisting of 1393 intensity values in the MCS and EMS-98 scale following a detailed appraisal of the original information. We finally discuss the results obtained in terms of intensity differences and macroseismic parameters resulting from the updated dataset. We also present an additional dataset of hydrological changes induced by the 1990 earthquake; these original and unpublished observations provide alternative estimates of the event magnitude based on environmental effects.

## 55 2 Data related to the Potenza 1990 earthquake

The May 5, 1990 seismic event with Mw 5.77 provoked a very large perception in southern Italy, causing light damage in a restricted area of Potenza province (southern Italy). The earthquake was localized at about 20 km depth (Azzara et al., 1993; Di Luccio et al., 2005; Castello et al., 2006; International Seismological Centre, 2025). Notwithstanding its moderate magnitude and limited damage to buildings, the 1990 event is considered the strongest recent earthquake of the eastern side of southern Apennines. The regional importance of the 1990 earthquake is highlighted in many studies, where it is considered one of the key events for shedding light on the seismotectonic of this sector of the southern Apennines. In this respect, the 1990 Potenza mainshock (and the following sequence) occurred on a strike-slip seismogenic structure at a depth well below 15 km (i.e., Azzara et al., 1993; Di Luccio et al., 2005; Frepoli et al., 2005; Boncio et al., 2007; Maggi et al., 2009). On the contrary, very few studies have focused on the macroseismic implications of this earthquake (Alessio et al., 1995; Tertulliani et al., 1992). As gathered in the Italian Archive of Historical Earthquake Data ASMI (Rovida et al., 2017; Rovida et al., 2025), two different macroseismic datasets related to the Potenza earthquake are available in literature (Fig. 1).

Figure 1: Intensity distribution in the MCS scale of the 5 May 1990 earthquake as in ASMI. MDP set by Tertulliani et al., (1992) with the name of the localities cited in the text (a) and Gasparini et al., (1991) (b). The black star represents the instrumental epicentre as reported in CPTI15.

The first (Tertulliani et al., 1992) is the report which describes the field activities performed by Istituto Nazionale di Geofisica (hereinafter ING) personnel in the aftermath of the event to evaluate the macroseismic effects in the epicentral area. The macroseismic field survey conducted after the event was limited to 14 settlements and lasted only a few days, not enough to visit all the localities of the epicentral area (Fig. 1a). However, the picture that was obtained by the investigators was considered, at the time, sufficient to constrain the damaged area. The intensity was assessed using the MCS scale with the maximum intensity equal to 7 in Pietragalla (Fig. 1a). In general, the visited localities showed a picture of minor, not widespread damage, mainly limited to cracks in walls and falling tiles in vulnerable buildings. Figure 2 shows some damage observed in Pietragalla and Cancellara (Basilicata region; see Fig. 1a) related to two different types of buildings.

Figure 2. Example of damage caused by the 1990 event in the village of Pietragalla (left) and Cancellara (right). Intensity 7 and 6-7 MCS from Tertulliani et al., (1992), respectively.

The second dataset is the list of localities affected by the earthquake collected in the Macroseismic Bulletin of Istituto Nazionale di Geofisica BMING (Gasparini et al., 1991). The Istituto Nazionale di Geofisica e Vulcanologia (INGV; ING before 2000) developed a standard methodology for collecting macroseismic data between 1978 and 2006. This methodology involved sending specific questionnaires based on the MCS scale to public territorial offices and military corps of a given locality (e.g., Corpo Forestale dello

https://doi.org/10.5194/egusphere-2025-5343 Preprint. Discussion started: 7 November 2025 © Author(s) 2025. CC BY 4.0 License.

Stato). The completed questionnaires were returned to ING and then automatically processed to obtain estimates of macroseismic intensity. Intensity estimation was derived from a statistical algorithm that used a simple weighted average technique of all available questionnaires for each locality (Gasparini et al., 1992). During this period, this methodology allowed the ING to collect a great amount of macroseismic information in order to assign intensity to localities potentially affected by a given seismic event. Intensity data were collected and made available in the BMING, which was the main source of macroseismic data for most of the medium-to-low energy earthquakes that occurred in Italy from 1980 to 2006. With this methodology, BMING collected macroseismic intensity data related to the May 1990 earthquake for 1375 localities, extending from the epicentre to areas where the earthquake was imperceptible (see Fig. 1b).

#### 3 Analysis of the pre-existing data

The comparison of the two available intensity distributions overwhelmingly shows the enormous diversity of the scenarios proposed by the authors: one is represented by reliable data but limited to a few locations around the epicentre (Tertulliani et al., 1992); the other extremely rich in number of data, even far away from the epicentre, but highly inhomogeneous in the distribution of intensity values (Gasparini et al., 1991). In particular, one aspect of the data coming from the BMING questionnaires is particularly surprising, namely the presence of localities, at distances far beyond 100 km from the epicentre, marked by intensity values similar to those of the epicentral area (assigned through direct survey) (see Fig. 1b). The poor reliability and heterogeneity of these data are even confirmed by the authors of the CPTI15, who do not provide macroseismic parameters of this event but only the instrumental ones. To understand the reasons behind this inconsistency in the data and then assess the room for correction and improvement of the basic data, we undertook an analysis of the datasets to identify major data biases. To this end, we compared the intensities documented in the BMING dataset with synthetic values derived from recent Intensity Prediction Equation (IPE) proposed by Gomez-Capera et al. (2024) and calibrated for the Italian territory. The intensity (I) at a given site is estimated as:

$$I = 2.86 - 0.0020 * D - 3.26 * Log(D) + 1.43 * Mw$$
 (1)

Equation (1) is a classical log-linear attenuation model that predicts macroseismic intensity as a function of moment magnitude (Mw) and hypocentral distance (D). The latter is estimated from the epicentral distance of a given site and a fixed pseudo-depth. We selected the relationship reported in Eq. (1) because it employs a fixed pseudo-focal depth of 16 km, which is similar to the instrumental depth of this event (i.e., about 20 km). The standard deviation of the model is 0.75 (for further details, see Gomez-Capera et al., 2024). Figure 3 compares the Macroseismic Data Points (MDPs) from the BMING dataset with the macroseismic intensity decay predicted by the GOM24 model. The largest discrepancies between the observed and predicted intensities are observed at the sites located between approximately 40 and 160 km of the epicentre, especially for intensities greater than 5 MCS. To understand the origin of this heterogeneity, we systematically checked all MDPs in the BMING dataset according to a priority scheme: (i) intensity values that differ by more than two standard deviations (i.e., ±1.50) from the GOM24 model's estimate; (ii) intensity values that differ by more than one standard deviation (i.e., ±0.75) and (iii) all other intensity values.

Figure 3. Intensity data provided by BMING dataset in comparison with the decay of macroseismic intensity as a function of the epicentral distance for the selected IPE. The thin and dotted lines represent the standard and two-standard deviation of the model, respectively.

Concerning the first two priority points, we reappraised all available sources related to the localities that display such intensity values, with a special focus on the re-reading of the original questionnaires. This review revealed that in many cases the answers given by the questionnaire compiler were contradictory or clearly exaggerated when compared to the effects found in other sources or in the field survey. In this respect, we highlighted potential overlap or cumulated effects from previous seismic events. We identified the 23 November 1980, Mw 6.8 Irpinia earthquake (<a href="https://emidius.mi.ingv.it/CPTI15-DBMI15/eq/19801123">https://emidius.mi.ingv.it/CPTI15-DBMI15/eq/19801123</a> 1834\_000, last access: 27 October 2025) and the 7 May 1984, Mw 5.86 lower Latium earthquake (<a href="https://emidius.mi.ingv.it/CPTI15-DBMI15/eq/19840507">https://emidius.mi.ingv.it/CPTI15-DBMI15/eq/19840507</a> 1749\_000, last access: 27 October 2025). Although their epicentres are distant, their extensive damage patterns partially overlap with those of 5 May 1990. Moreover, we also investigated possible overlapped effects due to the seismic sequence that occurred on 22 April 1990 in the Sannio area (<a href="https://emidius.mi.ingv.it/CPTI15-DBMI15/eq/19900422">https://emidius.mi.ingv.it/CPTI15-DBMI15/eq/19900422</a> 0945\_000, last access: 27 October 2025).

The overlap of the May 5, 1990 earthquake to these previous events may have contributed, on one hand, to confusing information between the effects (especially for the Sannio and Irpinia earthquakes), and on the other hand, it provided, for correspondents from locations already damaged in 1980 and 1984, an additional opportunity to emphasize the need for attention to damage that had not yet been repaired. This latter consideration derives from the careful and critical rereading of the macroseismic questionnaires. An example is represented in Figure 4. In detail, Fig. 4 reports the original BMINGV questionnaire related to one site (i.e., Apollosa; see Fig. 1a) located in the Campania region, 96 km NW of the CPTI15 epicentre. The correspondent who compiled the form clearly provides additional information highlighting that the damage caused by the 1980 Irpinia earthquake, still largely unrepaired, has been further exacerbated.

40 Figure 4. Example of the macroseismic questionnaire related to the Apollosa village, 96 km far from the epicentre. The questionnaire consists of three different sections, that refer to effects on people, damage, and effects on the environment. At the bottom of the form the Municipal engineer handwrote the note (highlighted in the figure) "worsening of damage caused by the 23/11/80 and not yet repaired".

The same kind of information can be inferred from several other questionnaires related to localities already hit by the 1980 earthquake. In particular, Figure 5 illustrates the geographical overlap between sites in the BMING dataset with an intensity greater than or equal to 6 MCS that already experienced significant damage ( $I \ge 6$  MCS) during the event that occurred on the 23 November 1980 according to Guidoboni et al. (2007). In addition, the intensity 6-7 MCS reported in the BMING dataset at the Venafro village, located more than 160 km from the 1990 earthquake's epicentre (see Fig. 1a), can be considered a possible overlap effect, given that the site was damaged by the 7 May 1984 earthquakes. As expected, most of those sites display intensities differing more than two standard deviations in comparison with the estimates of the GOM24 model (see Fig. 3).

Figure 5: Geographical distribution of the localities with I≥6 MCS (BMING dataset) that were also damaged by the 23 November 1980 Irpinia earthquake.

The occurrence at considerable distances of intensities higher than expected could also suggest the presence of local amplification phenomena. However, even though we cannot a priori exclude this possibility in isolated cases, we believe that the probability of

155 finding such a high number of localities exhibiting a site effect, fairly evenly distributed, at considerable distances from the epicentre, and due to a moderate magnitude earthquake, is very low. In summary, we can infer that disturbance factors may derive from previous earthquakes that occurred close to the area affected by a given earthquake, whose perception of the effects could have been enhanced and/or distorted (Cucci and Tertulliani, 2007; Joffe et al., 2013; Tertulliani et al, 2014; Becker et al., 2019). This is certainly the case with the 1980 Irpinia earthquake (Mw 6.9), which has long remained in the memory of the population due to the level of destruction it caused and the lengthy reconstruction work that followed.

As a final step, we checked the macroseismic questionnaires for all remaining localities. Based on this review, we assessed a new intensity value for each of these sites.

#### 4 Resulting dataset

Following the priority scheme outlined in Section 3, we assessed a total of 1,393 new intensity values, which were compiled from a careful review of all available macroseismic questionnaires and other sources (e.g., reports, newspapers), as detailed in Section 2. The resulting data compilation is provided in the Supplement (Table S1). Figure 6 shows the resulting intensity distributions on both the MCS and EMS-98 scales. In the area with the most significant effects, 18 localities experienced an intensity of 6 MCS or higher within 40 km of the epicentre. In contrast, an intensity of 6 EMS-98 or higher was assigned at seven sites within 27 km. The two datasets show the same maximum intensity of 6-7 assigned at four localities on the MCS scale (Fig. 6a) and at one site on the EMS-98 (Fig. 6b).

Figure 6. Macroseismic intensity distributions in the MCS (a) and EMS-98 scales (b) provided in this study.

Figure 7 illustrates the data distribution of the new dataset across the MCS and EMS-98 scales. While the frequency of the data is very similar for all intensity classes, significant differences emerge at intensities 5, 5-6, and 6. Figure 7 shows that the number of MDPs is 6 on the EMS-98 scale, compared to 14 on the MCS scale. A similar pattern is observed for intensity 5-6, with 22 MDPs for EMS-98 versus 54 for MCS. Furthermore, differences between the two scales are observed for intensity 5, with the increase of 42 intensity data on the EMS-98 with respect to the MCS scale. The differences in the frequency of data across these intensity

classes can be attributed to the different diagnostics criteria employed by the two scales. These discrepancies have been thoroughly examined and discussed in recent studies (Del Mese et al., 2023; Tertulliani et al., 2025).

Figure 7. Comparison of the new dataset on the MCS (red bars) and EMS-98 (blue bars) scales.

Figure 8 shows the comparison between the new data and those coming from the BMING dataset. In detail, Figure 8a reveals a significant decrease in the number of high intensities (i.e.,  $\geq 5\text{-}6$  MCS) in the new dataset, alongside an increase in data for lower intensities (i.e., 4 and for 4-5 MCS). In addition, Figure 8b demonstrates that the reduction in the number of intensities greater than or equal to 5 is most pronounced for sites located within 150 km of the epicentre. As previously discussed, this may be attributed to an overestimation of the effects resulting from the original macroseismic questionnaires. Subsequent re-reading of the sources revealed that reports of damage, even extensive damage, were not consistent for a considerable number of localities after cross-referencing all the sources. Consequently, the intensity degree was downgraded for many of these localities.

Figure 8. Comparison between the MDPs of the new dataset (red) and the BMING dataset (green), both on the MCS scale as a function of number of MDPs (a) and epicentral distance (b).

The spatial distribution of differences between the new intensity assessment and the BMING dataset is represented in Figure 9. Although 78% of the new intensity assessments are equal to the previous dataset, the trend indicates a decrease in intensity when compared to the BMING dataset within approximately 150 km of the epicentre (see Fig. 1b). In particular, about 10% of the data show a decrease of half a degree and around 4% show a decrease of one full degree. In two localities, the BMING intensity was found to be overestimated by two full degrees. On the contrary, in a very limited number of cases (14 sites), the new intensities are higher than those of BMING. This re-evaluation also led to several updates of the dataset. For 22 sites, information based on questionnaires and other sources was deemed unreliable, leading us to not assess an intensity value. In this case, we assigned it as

"Not Classified" (NC). Moreover, further sources are identified for 18 sites not previously included in the BMING dataset, for which new intensity values have been assessed (Fig. 9). In comparison with the 14 intensities evaluated by Tertulliani et al. (1992) via macroseismic field survey (see Fig.1a and Section 2), the new assessment is half a degree lower at 6 out of 14 sites and one degree lower in 1 locality.

Figure 9. Differences in terms of intensity values between the new data and those coming from BMING dataset. Removed (black crosses) and added data (green stars) are also represented.

Finally, we calculated the macroseismic epicentre and magnitude using both the MCS and EMS-98 intensity datasets. We used the same methodology used for the CPTI15 Catalogue: BOXER (Gasperini et al., 1999; 2010). The results are summarized in Table 1. The estimated macroseismic epicentres differ from the CPTI15 instrumental one by 7.3 km using MCS data, and 16.8 km using 210 EMS-98 data. As expected, the calculated macroseismic magnitudes are significantly lower than the instrumental magnitude provided in the Italian Catalogue. This is due to the methodology used to compute magnitude (i.e., BOXER), which bases its magnitude calculation on epicentral intensity (10) and the isoseismal areas. The former is based on the number of data with maximum intensity and the latter is computed considering the average distance between the macroseismic epicentre and each locality for a given intensity class (see Gasperini et al., 2010 for further details). In this case, the estimated I0 is moderate (i.e., 6-7 MCS and 6 EMS-98) and the number of points with intensity 5-6, 6, and 6-7 is fairly limited, resulting in smaller average isoseismal areas and, consequently, a lower magnitude estimate. However, these relative low effects are likely due to the depth of this event, estimated instrumentally to be around 20 km (Castello et al., 2006; Di Luccio et al., 2005; International Seismological Centre, 2025). To investigate this further, we also estimated the source depth using our macroseismic data with the method developed by Sbarra et al. (2023) for historical earthquakes. The analysis yielded a depth between 30 and 40 km, confirming that the May 5, 1990 earthquake was not a shallow event (see Sbarra et al., 2023 for details and reliability of the method).

Table 1. Instrumental parameters as reported in the CPTI15 and macroseismic parameters estimated with the data in the MCS and EMS-98 scale.

| Ref                 | Lat    | Lon    | Mw   | Mw Unc | 10  |
|---------------------|--------|--------|------|--------|-----|
| This Study [EMS-98] | 40.703 | 15.935 | 5.05 | 0.04   | 6   |
| This Study [MCS]    | 40.694 | 15.805 | 5.19 | 0.04   | 6-7 |
| CPTI15              | 40.738 | 15.741 | 5.77 | 0.10   | -   |

240

245

#### 5 Hydrological changes induced by the Potenza earthquake

The ING macroseismic questionnaires (see Section 2) also contained questions about environmental effects, which include hydrological anomalies. Seismically-induced hydrological changes are among the most outstanding effects produced by earthquakes on the environment. They may include increases or decreases in streamflow and groundwater levels, the appearance or disappearance of springs, changes in spring discharge, turbidity, liquefaction, and variations in the chemical and physical properties of water. The careful review of the macroseismic questionnaires allowed us to identify 69 instances of hydrological change across 56 different locations (Fig. 10) generated by the 1990 Potenza earthquake.

Figure 10. Map of the hydrological effects induced by the 1990 Potenza earthquake. Numbers in the circles correspond to the observations listed in Appendix A. A black star indicates the position of the epicentre of the 1990 seismic event from the CPTI15 (Rovida et al., 2022). Observations no. 15, 19 and 50 are located outside the limits of the map.

- Most reports described increased turbidity in ponds and streams, as well as variations in spring discharge and well water levels. Reports of liquefaction were less common. In addition, two further observations of hydrological changes in streams were obtained from the Hydrological Annals of the National Hydrographic Service. In total, we collected 71 hydrological observations associated with the 1990 event, distributed across 58 sites (see Appendix A).
  - The average epicentral distance of these observations is approximately 52 kilometers-slightly larger than that reported for similar seismic sequences in southern Italy (Cucci et al., 2024, 2025).
  - To further constrain the size of the 1990 seismic event, we estimated its magnitude using empirical relationships that relate earthquake magnitude (M) to the maximum distance to which specific hydrological effects are observed. Several studies have proposed such relationships for liquefaction (Galli, 2000; Montgomery and Manga, 2003; Pirrotta et al., 2007; Hu, 2023) as well as for streamflow responses (Montgomery and Manga, 2003). Our results, illustrated in Figure 11 and Table 2, indicate that magnitudes inferred from liquefaction effects range between M 4.9 and 5.7, whereas estimates based on streamflow responses suggest a magnitude of M 5.2. These independent approaches provide complementary constraints on the event's overall size.

Figure 11. Distance from epicentre versus earthquake magnitude for locations that exhibited the farthest effects of liquefaction (diamonds) or of streamflow response (circle) induced by the 1990 earthquake (see Appendix A). Empirical relationships from several authors show that magnitudes inferred from liquefaction effects range between M 4.9 and 5.7, whereas estimates based on streamflow responses suggest a magnitude of M 5.2. Abbreviations: HU23, Hu 2023; MM03, Montgomery and Manga, 2003; PI07, Pirrotta et al., 2007; GA00, Galli 2000.

Table 2. Earthquake magnitudes from hydrological effects (abbreviations as in Figure 11).  $M_{liq}$  and  $M_{str}$  are the magnitude calculated on liquefaction and streamflow effects respectively.

| Ref  | $M_{liq}$ (12 km) | $M_{str}$ (165 km) |
|------|-------------------|--------------------|
| MM03 | 5.6               | 5.2                |
| GA00 | 4.9               | -                  |
| HU23 | 5.7               | -                  |
| PI07 | 5.2               | -                  |

#### 255 6 Discussion and conclusions

In this work, we present a revised and comprehensive macroseismic dataset of the 5 May 1990 earthquake that occurred in southern Italy, comprising 1393 MDPs with an assessed intensity on both MCS and EMS-98 scales (Table S1). To achieve this goal, we considered all available sources (see Section 2) and analyzed the pre-existing datasets applying a priority scheme to re-evaluate each MDP (Section 3). This analysis revealed several contradictions and clear inaccuracies in the former intensity estimates, allowing the identification of misinterpreted effects at some sites. We suggested that these effects are likely caused by the partial overlapping of damage patterns from previous strong earthquakes that occurred in the same area, such as the 23 November 1980 earthquake. As a result, we first corrected biased data in the BMING dataset by cross-referencing them with other sources, such as field survey reports, newspapers or other accounts, to enrich the overall dataset. The new dataset shows a general trend of decreased intensity values, with approximately 10% of the data that are lower by half a degree and 4% by one full degree compared to those provided by the BMING dataset. However, in a few cases (i.e., 14 localities), the new intensity is higher than the previous one. Furthermore, we evaluated an intensity value at 18 new sites not previously included in the BMING dataset, and we removed unreliable data. We compute the macroseismic parameters using the data expressed on both scales. While the resulting epicentres are consistent with the CPTI15 instrumental location (differing by 7.3 km and 16.8 km for MCS and EMS-98 data, respectively), the macroseismic magnitudes (i.e., 5.19 with MCS; 5.05 with EMS-98) are significantly lower than the instrumental one. These differences are a direct consequence of our revised assessment, which substantially reduced the number of higher intensity values, particularly for intensities 5-6, 6, and 6-7 MCS (see Fig. 8 and Section 4).

A significant contribution to the knowledge of the 1990 earthquake, and in particular of its size, is provided by the observations of hydrological changes induced by the event. We produce 71 unpublished instances of hydrological variations (Appendix A) that represent a remarkable set of data for a moderate magnitude earthquake. The empirical relationships that relate the maximum

distance of occurrence of environmental effects (liquefaction, streamflow response) to the energy of an earthquake provide an independent constraint to the 1990 event overall size and propose magnitude values downsized as much as the macroseismic magnitude. In addition, the downsizing of the macroseismic magnitude can be partly accounted for by the remarkable depth of the 1990 event. This last evidence is also supported by the independent macroseismic approach for estimating the depth proposed by Sbarra et al. (2023). As a matter of fact, most of the investigators agree on the hypothesis that this section of the southern Apennines is characterized by extensional seismicity affecting the upper 15 km of the crust, and by deeper strike-slip faults cutting the crystalline basement of the chain. Under this view, the 1990 earthquake would represent an example of the nucleation of strike-slip earthquakes at deep crustal levels, in close proximity with shallow normal faulting earthquakes like the Mw 4.0 event that occurred in March 2025 (https://terremoti.ingv.it/event/41973812, last access: 27 October 2025).

This work significantly improves the knowledge of the seismic effects related to the May 1990 earthquake in southern Italy. The
resulting dataset can be integrated into future versions of the Italian Macroseismic Database - DBMI. Furthermore, this dataset can
now be used for a wide range of seismological purposes, including calibrating methodologies to derive earthquake parameters,
developing accurate intensity prediction equations and ground-motion-to-intensity conversion equations, and assessing seismic
hazard using a site-specific approach.

#### Appendix A

290 This section contains the list of the hydrological changes induced by the 1990 Potenza earthquake described in Section 5.

Table A1. N: Record number; Lat: Latitude (WGS84); Lon: Longitude (WGS84); Place Name: Locality Name; Dist.: Epicentral Distance (km); Observation: Type of hydrological effect; Notes: Source of the data (A: macroseismic questionnaire; B: macroseismic questionnaire from Corpo Forestale dello Stato; C: Hydrological Annals).

| N  | Lat    | Lon    | Place Name                | Dist.  | Observation                                                                             | Notes |
|----|--------|--------|---------------------------|--------|-----------------------------------------------------------------------------------------|-------|
| 1  | 41.158 | 15.334 | Accadia                   | 57.87  | decrease of spring discharge and/or of well level                                       | A     |
| 2  | 40.516 | 15.924 | Anzi                      | 29.12  | turbid water from pond, ditch, stream                                                   | A     |
| 3  | 41.093 | 14.701 | Apollosa                  | 95.89  | decrease of spring discharge and/or of well level                                       | A     |
| 4  | 40.306 | 16.066 | Armento                   | 55.34  | decrease of spring discharge and/or of well level;<br>stream disappeared for eight days | A     |
| 5  | 40.877 | 15.653 | Atella                    | 17.14  | turbid water from pond, ditch, stream                                                   | A     |
| 6  | 40.650 | 15.512 | Balvano                   | 21.64  | turbid water from pond, ditch, stream                                                   | A     |
| 7  | 40.681 | 15.591 | Baragiano                 | 14.14  | decrease of spring discharge and/or of well level                                       | Α     |
| 8  | 41.393 | 14.973 | Baselice                  | 97.21  | new spring                                                                              | A     |
| 9  | 41.102 | 15.004 | Bonito                    | 73.98  | increase of spring discharge and/or of well level                                       | A     |
| 10 | 40.478 | 15.629 | Brienza                   | 30.42  | turbid water from pond, ditch, stream                                                   | A     |
| 11 | 40.885 | 15.363 | Cairano scalo             | 35.78  | increase of streamflow discharge                                                        | С     |
| 12 | 40.475 | 15.849 | Calvello                  | 30.63  | turbid water from pond, ditch, stream                                                   | A     |
| 13 | 40.565 | 16.072 | Campomaggiore             | 33.91  | turbid water from pond, ditch, stream                                                   | A     |
| 14 | 40.731 | 15.923 | Cancellara (loc. Bufate)  | 15.35  | turbid water from pond, ditch, stream                                                   | В     |
| 15 | 41.106 | 14.214 | Capua                     | 134.66 | turbid water from pond, ditch, stream                                                   | A     |
| 16 | 40.150 | 15.620 | Casaletto Spartano        | 66.18  | decrease of spring discharge and/or of well level                                       | A     |
| 17 | 41.297 | 15.085 | Castelfranco in Miscano   | 83.02  | turbid water from pond, ditch, stream                                                   | A     |
| 18 | 41.297 | 15.085 | Castelfranco in Miscano   | 83.02  | new spring                                                                              | A     |
| 19 | 41.866 | 14.451 | Castiglione Messer Marino | 165.36 | decrease of spring discharge and/or of well level                                       | A     |
| 20 | 40.809 | 15.708 | Filiano                   | 8.37   | turbid water from pond, ditch, stream                                                   | A     |
| 21 | 40.809 | 15.708 | Filiano                   | 8.37   | increase of spring discharge and/or of well level                                       | A     |
| 22 | 40.859 | 15.855 | Forenza                   | 16.53  | turbid water from pond, ditch, stream                                                   | В     |
| 23 | 40.859 | 15.855 | Forenza                   | 16.53  | decrease of spring discharge and/or of well level                                       | В     |
| 24 | 40.859 | 15.855 | Forenza                   | 16.53  | increase of spring discharge and/or of well level                                       | В     |
| 25 | 40.459 | 15.971 | Laurenzana                | 36.60  | decrease of spring discharge and/or of well level                                       | A     |
| 26 | 40.459 | 15.971 | Laurenzana                | 36.60  | increase of spring discharge and/or of well level                                       | A     |

| 27  | 40 421 | 15 725 | Manaira Marana             | 25.25  | 41.146                                             | D |
|-----|--------|--------|----------------------------|--------|----------------------------------------------------|---|
| 27  | 40.421 | 15.735 | Marsico Nuovo              | 35.25  | turbid water from pond, ditch, stream              | В |
| 28  | 40.376 | 15.824 | Marsicovetere              | 40.86  | turbid water from pond, ditch, stream              | В |
| 29  | 40.342 | 15.542 | Monte San Giacomo          | 47.13  | new spring                                         | A |
| 30  | 40.679 | 14.946 | Montecorvino Pugliano      | 67.33  | turbid water from pond, ditch, stream              | A |
| 31  | 40.694 | 14.977 | Montecorvino Rovella       | 64.58  | increase of spring discharge and/or of well level  | A |
| 32  | 40.552 | 16.667 | Montescaglioso             | 80.82  | turbid water from pond, ditch, stream              | A |
| 33  | 40.099 | 16.483 | Nocara                     | 94.84  | turbid water from pond, ditch, stream              | A |
| 34  | 40.099 | 16.483 | Nocara                     | 94.84  | decrease of spring discharge and/or of well level  | A |
| 35  | 40.148 | 16.540 | Nova Siri                  | 94.21  | turbid water from pond, ditch, stream              | A |
| 36  | 40.340 | 15.659 | Padula                     | 44.79  | turbid water from spring                           | A |
| 37  | 40.930 | 15.986 | Palazzo San Gervasio       | 29.68  | increase of well level                             | A |
| 38  | 41.036 | 14.580 | Paolisi                    | 103.07 | turbid water from pond, ditch, stream              | Α |
| 39  | 40.377 | 15.732 | Paterno                    | 40.15  | turbid water from pond, ditch, stream              | В |
| 40  | 40.542 | 15.450 | Pertosa                    | 32.83  | turbid water from pond, ditch, stream              | A |
| 41  | 40.747 | 15.881 | Pietragalla                | 11.84  | turbid water from pond, ditch, stream              | A |
| 42  | 40.735 | 15.855 | Pietragalla                | 11.84  | liquefaction                                       | A |
| 43  | 40.638 | 15.802 | Potenza                    | 12.25  | turbid water from pond, ditch, stream              | A |
| 44  | 41.223 | 14.450 | Puglianello                | 121.05 | turbid water from pond, ditch, stream              | A |
| 45  | 40.976 | 15.675 | Rapolla                    | 27.04  | turbid water from pond, ditch, stream              | A |
| 46  | 40.976 | 15.675 | Rapolla                    | 27.04  | decrease of spring discharge and/or of well level  | A |
| 47  | 40.972 | 14.561 | Roccarainola               | 102.60 | turbid water from pond, ditch, stream              | A |
| 48  | 40.591 | 15.460 | Salvitelle                 | 28.79  | turbid water from pond, ditch, stream              | Α |
| 49  | 40.190 | 16.075 | San Chirico Raparo         | 67.17  | turbid water from pond, ditch, stream              | Α |
| 50  | 40.987 | 14.174 | San Marcellino             | 134.65 | turbid water from pond, ditch, stream              | Α |
| 51  | 41.236 | 14.499 | San Salvatore Telesino     | 118.04 | decrease of spring discharge and/or of well level  | A |
| 52  | 40.545 | 15.559 | Sant'Angelo le Fratte      | 26.39  | turbid water from pond, ditch, stream              | A |
| 53  | 40.870 | 14.876 | Santa Lucia di Serino      | 74.27  | turbid water from pond, ditch, stream              | A |
| 54  | 40.870 | 14.876 | Santa Lucia di Serino      | 74.27  | increase of spring discharge and/or of well level  | A |
| 55  | 40.336 | 15.561 | Sassano                    | 47.22  | turbid water from pond, ditch, stream              | A |
| 56  | 40.336 | 15.561 | Sassano                    | 47.22  | increase of spring discharge and/or of well level  | Α |
| 57  | 40.488 | 15.677 | Sasso di Castalda          | 28.32  | increase of spring discharge and/or of well level  | A |
| 58  | 40.543 | 15.639 | Satriano di Lucania        | 23.33  | turbid water from pond, ditch, stream              | Α |
| 59  | 40.801 | 14.693 | Siano                      | 88.53  | decrease of spring discharge and/or of well level; | A |
|     |        |        |                            |        | observed at three different springs                |   |
| 60  | 40.582 | 15.675 | Tito                       | 18.22  | turbid water from pond, ditch, stream              | В |
| 61  | 40.696 | 16.019 | Tolve                      | 23.89  | turbid water from pond, ditch, stream              | A |
| 62  | 40.696 | 16.019 | Tolve                      | 23.89  | increase of spring discharge and/or of well level  | A |
| 63  | 40.696 | 16.019 | Tolve                      | 23.89  | new spring                                         | A |
| 64  | 41.189 | 14.680 | Torrecuso                  | 102.23 | turbid water from pond, ditch, stream              | A |
| 65  | 41.189 | 14.680 | Torrecuso                  | 102.23 | decrease of spring discharge and/or of well level  | A |
| 66  | 40.315 | 15.790 | Tramutola                  | 47.22  | turbid water from pond, ditch, stream              | A |
| 67  | 41.048 | 15.234 | Trevico                    | 54.81  | turbid water from pond, ditch, stream              | A |
| 68  | 40.580 | 15.990 | Trivigno                   | 27.38  | decrease of spring discharge and/or of well level  | В |
| 69  | 40.666 | 15.921 | Vaglio Basilicata          | 17.16  | increase of spring discharge                       | A |
| 70  | 40.961 | 15.818 | Venosa                     | 25.63  | turbid water from pond, ditch, stream              | A |
| 71  | 40.980 | 15.843 | Venosa ponte ferroviario   | 28.23  | decrease of streamflow discharge                   | C |
| , 1 | 10.700 | 12.072 | . those points ferroviario | 20.23  | accitable of bacamino,, albeitaige                 | ~ |

# 295 Data availability

The dataset provided by Tertulliani et al. (1992) and Gasparini et al. (1991) were downloaded from the Italian Archive of Historical Earthquake Data - ASMI (https://emidius.mi.ingv.it/ASMI/index\_en.php, last access: 27 October 2025).

# **Supplement**

The supplementary material for this article contains the new macroseismic distribution related to the 5 May 1990 earthquake in 300 Southern Italy (Table S1).

#### **Author contribution**

All authors contributed equally to the conceiving and writing of the work. AA built the macroseismic dataset, LC built the hydrological dataset.

## Competing interests

The authors declare that they have no conflict of interest.

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
