# Peer review of "A thorough review of the May 5, 1990 earthquake (southern Italy): constraints from macroseismology and insights from hydrology"

_EGUsphere, 2025_

## Author Comment (AC1)

We would like to thank the anonymous reviewer for their suggestions. We responded to the main comments point by point and considered almost all of the minor and specific suggestions. We have also modified Figures 4 and 10 as requested.

**General comments**

The paper brings a new insight into consequences of 5 May 1990 Potenza earthquake. The new analysis of the existing datasets, with the addition of new primary sources, shows an updated picture of this damaging event. Bringing hydrological effects into the picture allows us to have an additional insight into the earthquake effects in the wider epicentral area.

Please spell-check the text carefully, using UK English for all used terms.

The date format: the use of the scientific date format is recommended (DD Month YEAR) in the entire paper.

We have modified the data and we have carefully checked the terms used.

**Specific comments**

Site vs. locality: I have been researching the use of both terms, and it seems that there is no consensus which one is more proper for naming an inhabited place. Several sciences (e.g. archaeology, palaeontology etc) use these terms not as synonyms, but as a definition of the size of the examined area. In my head locality relates to man-built habitations, and a site can be just anything, from *a camp site* to *intended **site** for a new shopping mall*; but it's not necessarily true. However, I suggest that the authors read the text again and unify the use of the chosen term.

We agree with this comment. No specific definition is provided to differentiate between the two terms. For this reason, we have adopted the definition of 'locality' from the latest version of the Italian Macroseismic Database (DBMI15) (https://emidius.mi.ingv.it/CPTI15-DBMI15/description_DBMI15_en.htm), replacing all the terms 'site' with 'locality' in the manuscript.

The title of the paper: as the earthquake in focus is being referenced as Potenza earthquake throughout the text, this name should be used in the title, too. Consider changing it into " A thorough review of the 5 May 1990 Potenza (Southern Italy) earthquake: constraints from macroseismology and insights from hydrology"

We have modified the title as suggested.

Epicentral intensity I0 should be written with zero as subscript (see https://gfzpublic.gfz.de/rest/items/item_4011_4/component/file_4012/content).

Ok, modified.

When seismologists talk about all sorts of things that can be felt and/or leave a trace on the seismograms, they are named as events. However, when we talk of a known earthquake, to avoid any possible confusion, it's called an earthquake and not an event.

We used the term "event" to avoid repetition in the text. Several recent articles used "event" as a synonym for "earthquake", even though this is known (i.e., D'Amico et al., 2025; Sbarra et al., 2023; Vannucci, 2024).

Line 8 and further in the text: just out of curiosity, is the precision with which intensities are evaluated enough to allow us expressing the Mw with two decimal places?

The BOXER method computes the macroseismic magnitude by combining epicentral intensity and the isoseismal areas of each intensity class using an intensity prediction equation. Macroseismic magnitude (M) is computed as the weighted average of values obtained independently from each intensity class through the equation. For this reason, it is expressed with two decimal places (see Gasperini et al., 2010, for further details).

Line 13: "intensity values". It should be clarified, here as well in the rest of the text, that the authors are talking about macroseismic data points (MDPs). Please check and correct all the places it applies to (line 50 and forward). It is a good place to explain what MDPs are, as the abbreviation is used already in Fig. 1 and explained only later in the text.

In line 11 of the track changes manuscript, we simply refer to the numerical value of macroseismic intensity. We have added "macroseismic" to this part to clarify this point. In general, a macroseismic data point (MDP) represents the intensity value associated with a specific locality with its name and coordinates, referred to a given event. We have carefully checked the use of "MDP" and "intensity value" terms, adding the term 'MDP' where we refer to the final dataset.

Line 56: consider replacing "provoked a very large perception" with "was widely felt"

Ok, modified.

Figure 1: MDP – already mentioned. Data points with Imax are not visible in the maps. Consider adding the number of Imax data points and the Imax intensity in the caption.

We have included the number of localities and their Imax for both studies in the caption of Figure 1.

Line 84: Sentence starting with "This methodology involved sending…" should be translated from Italian in a more precise way.

Ok, modified.

Line 112: abbreviation GOM24 is not explained

Ok, we have added the explanation of GOM24.

Figure 4: absolutely not readable, at least in the pdf I am working with.

Consider changing "damage and effects on the environment" to "buildings and nature".

We have created a clearer version of Figure 4.

Line 146: Guidoboni et al. (2007) not in the References

Ok, added.

Line 164: Consider replacing "we assessed a total of 1393 new intensity values, which were compiled from a careful" with "a total of 1393 MDPs were assessed, as a result of a careful".

Ok, modified.

Figure 6. Caption should read "distributions for 5 May 1990 Potenza earthquake in the…" or something similar.

Ok, modified.

Line 179: The last sentence needs a bit more details, at least 1-2 sentences describing very shortly what those 2 papers are about.

We have added a sentence at lines 192-194 of the track changes manuscript to provide a clearer description of the cited authors' perspective.

Line 195: "In two localities" – name them.

We have added the name of the two localities and we have reported their location in Fig.1a.

Line 202: consider replacing "the new assessment is" with "the intensities assessed in this study are"

Ok, modified.

Table 1. Consider adding another column with Imax.

We have added an Imax column to Table 1.

Line 225: instead of "see Section 2" there should be "see Figure 4", I presume?

Yes, modified.

Figure 10: the background of the map is too dark and does not allow the tiny font to be readable

We have created a new version of Figure 10 with a light grey background.

Line 237: Hydrological Annals – not in the References

We have added "Hydrological Annals" to the "Data availability" section of the manuscript.

Line 241: consider replacing "To further constrain" with "In order to constrain further"

Ok, modified.

Line 254: replace "effects respectively" with "effects, respectively".

Ok, modified.

Line 255: the section gives good overview of the paper, but there's not much discussion.

We have changed the name of the chapter to "Conclusions".

Table S1: could you, instead of the category NC, differentiate the effects more and use the descriptive terms F (felt), damage (D) etc, as suggested in https://emidius.eu/MIDOP/manual/input_data_preparation/input_data_table_formats.php ?

We have followed the guidelines reported in the MIDOP manual and opted to assign the descriptive code NC (not classified) rather than "Felt" or "Damage". As stated in Lines 213-2115 of the track changes manuscript, this is because for some localities, there was insufficient information in the source to describe the effects for some localities. This definition is also given in the table of the MIDOP manual table that lists "Unconventional macroseismic descriptive codes".

Line 267: rephrase the sentence starting with "We compute". Are you talking about the intensity assessment of the MDPs or the calculation of the focal parameters?

We estimated the macroseismic parameters (see Table 1) using MDPs on the MCS and EMS-98 scales. We have added a reference to Table 1.

Line 283: "in March 2025" – date? Where?

We have added the details of this earthquake.

**Appendix A**

WGS84 not explained

Lat – add °N

Lon – add °E

A: add ING

We have considered all of these suggestions.

**Technical corrections**

Figure 4: absolutely not readable, at least in the pdf I am working with.

Figure 10: the background of the map is too dark and does not allow the tiny font to be readable

Line 239: replace "kilometers-slightly" with "kilometres – slightly

We took into account all the technical corrections suggested.

**References**

D'Amico, S., Tuvè, T. & Mantovani, A. New relationships between macroseismic intensity and local magnitude for the volcanic region of Mt. Etna (Italy). J Seismol 29, 305–315 (2025). https://doi.org/10.1007/s10950-024-10274-9

Gasperini, P., Vannucci, G., Tripone, D., and Boschi, E.: The Location and Sizing of Historical Earthquakes Using the Attenuation of Macroseismic Intensity with Distance, B. Seismol. Soc. Am., 100, 2035–2066, https://doi.org/1785/0120090330 , 2010.

Sbarra, P., Burrato, P., De Rubeis, V., Tosi, P., Valensise, G., Vallone, R., and Vannoli, P.: Inferring the depth and magnitude of pre-instrumental earthquakes from intensity attenuation curves, Nat. Hazards Earth Syst. Sci., 23, 1007–1028, https://doi.org/10.5194/nhess-23-1007-2023 , 2023.

Vannucci, G.: Analysis of the macroseismic cumulative damage in the seismic sequences in Italy. Bull Earthquake Eng 23, 759–778 (2025), https://doi.org/10.1007/s10518-024-02073-x

---

## Author Comment (AC2)

We would like to thank Javier Fernández-Fraile for his valuable suggestions and for appreciating our work. We took most of the comments into consideration.

The following is a point-by-point response.

**A thorough review of the (southern Italy): constraints from macroseismology and insights from hydrology**

Andrea Antonucci, Corrado Castellano, Luigi Cucci, Andrea Tertulliani

**General comments**

The authors present an exhaustive review of all the macroseismic information concerning May 5, 1990 earthquake in southern Italy, analyzing and comparing previous studies with their results. They have reevaluated original information sources which are crucial to obtain accurate results, of the seismic intensity, the hypocenter and hydrological effects.

Sometimes macroseismic studies are not valued enough, but having a homogeneous seismic catalog is fundamental to begin any other seismic studies, such as seismic hazard studies or study the seismicity of any area.

Therefore, these results improve the seismic knowledge of this area of Italy.

In my opinion, in general terms, this manuscript (MS) is very well done, very rigorous and with a lot of details, that has allowed me to appreciate the amount of work and effort required to develop this study. Figures and tables are very clarifying and well selected.

The structure of the MS is very well.

The language of the MS isn't evaluated because it's not the native language of the reviewer.

All the recommendations that I have included here are mere suggestions that I think that can improve the explanations and comprehension of the paper, but it is up to the author to accept them or not.

**Particular comments of the MS:**

**Line 38-39:** You should mention that this problem has also been faced in other parts of the world. For example, in Spain, the recent study Fernandez-Fraile et al., (2025) makes a revision of the first part of the 20th century earthquakes, reevaluating all the contemporary sources.

Fernández-Fraile, J., Mattesini, M. & Buforn, E. Re-Evaluation of the Earthquake Catalog for Spain Using the EMS-98 Scale for the Period 1900–1962. Pure Appl. Geophys. 182, 1237–1261 (2025). https://doi.org/10.1007/s00024-024-03461-9

We agree with this comment. We neglected to mention this interesting work in the manuscript. We apologise!

**Line 45:** the "consistent criteria" are explained in the definition of EMS-98 itself. The criteria of Tertulliani et al., 2025 are particular for that work.

We are in full agreement. As stated in Line 46 of the track changes manuscript, this sentence focuses on Italian earthquakes that present incoherent or inconsistent data. Tertulliani et al. (2025) are an example of such events.

**Line 50:** Can you briefly indicate here the weaknesses of previous studies?

We have added a brief example of the weaknesses of previous studies (line 52 of the track changes manuscript) according to Section 3.

**Line 60**: "is highlighted in many studies" Indicate here some of them.

We have cited at line 63-64 of the track changes manuscript the articles by Azzara et al., 1993; Di Luccio et al., 2005; Frepoli et al., 2005; Boncio et al., 2007 and, Maggi et al., 2009.

**Line 99**: The intensity distribution may follow a pattern like the one you propose in line 106, but it will also be necessary to check whether the presence of high-intensity MDPs far from the epicenter is due to site effects or local amplifications. This cannot be ruled out without prior analysis. I have noticed that you mention it later (line 153), but I think you should mention it here.

We fully agree. This comparison of documented and synthetic intensities allowed us to identify potential biases in the intensity data, which could be due to errors in the intensity estimation or possible site effects. We have modified a sentence at Line 107-108 of the track changes manuscript.

**Line 111**: A complementary way to compare the MDPs with the model (the IPE proposed by Gomez Capera et al., 2024) is to study de residuals (differences between the model and the MDPs) and represent it with histograms or something similar.

Yes, in our work, we simply used the synthetic intensity to establish a systematic checking process for the macroseismic questionnaires, starting from those with significant differences between observed and calculated intensities (lines 120-121). However, an in-depth analysis of residuals is beyond the scope of this research.

**Line 122:** I miss examples of these exaggerations. I would include information literally taken from the questionnaires to support your conclusions.

As stated in line 140 and Figure 4, we provided an example in the text relating to the Apollosa site, which is located more than 90 km from the epicentre of the earthquake under study. Upon reading the questionnaires and comparing the information provided with other sources, we found exaggeration in the intensity estimate even at the Capodrise and Pannarano sites, which are located around 100 km from the epicentre. We have added a sentence to lines 143-147 indicating these two examples. The location of these sites is also shown in Figure 1a

**Line 193 and figure 9:** It could be interesting to represent the epicenters of 1980 and 1984 in the same figure, to study if the distribution of MDP with higher differences (between this re-evaluation and BMING dataset) are related to the previous earthquakes.

We highlighted potential overlap or cumulative effects from previous seismic events, such as the 1980 Irpinia earthquake and the 1984 Sannio earthquake, as the primary cause of the high intensity of the BMING dataset. Based on this, we would prefer to show the epicentres of these two earthquakes in Figure 5 rather than in Figure 9.

**Line 208:** Maybe a map with the information from Table 1 could be interesting.

We agree with the suggestion. To reduce the number of figures, we have decided to represent the epicentre of the final dataset in Figure 6, since this is strictly correlated with the two macroseismic distributions.

**Line 220:** Furthermore, the intensity distribution supports the conclusion that it is not a shallow focus. If it were very shallow, the earthquake would have affected the epicentral area and attenuated much more rapidly than an earthquake with a hypocenter at 30-40 km.

We fully agree with this comment. We have added a sentence at lines 235-267 of the track changes manuscript.

**Table 1:** You can include favorite depths in this table.

Sbarra et al. clearly state that uncertainty increases with depth, resulting in wider error ranges for depths greater than 35 km. For this reason, we prefer to present the range of computed depths in the text (Line 234) rather than including a single value in the Table.